# Interventions to reduce social isolation and loneliness during COVID-19 physical distancing measures: A rapid systematic review

Christopher Y. K. Williams[1]*, Adam T. Townson[1], Milan Kapur[1], Alice F. Ferreira[1], Rebecca Nunn[1], Julieta Galante[2,3], Veronica Phillips[4], Sarah Gentry[5], Juliet A. Usher-Smith[6]

1 University of Cambridge School of Clinical Medicine, Cambridge, United Kingdom, 2 Department of Psychiatry, University of Cambridge, Cambridge, United Kingdom, 3 National Institute for Health Research Applied Research Collaboration East of England, England, United Kingdom, 4 Medical Library, University of Cambridge, Cambridge, United Kingdom, 5 Norwich Medical School, University of East Anglia, Norwich, United Kingdom, 6 The Primary Care Unit, Department of Public Health and Primary Care, University of Cambridge School of Clinical Medicine, Cambridge, United Kingdom

* cykw2@doctors.org.uk

**Data Availability Statement:** All relevant data are within the paper and its Supporting Information files.

## Abstract

### Background

A significant proportion of the worldwide population is at risk of social isolation and loneliness as a result of the COVID-19 pandemic. We aimed to identify effective interventions to reduce social isolation and loneliness that are compatible with COVID-19 shielding and social distancing measures.

### Methods and findings

In this rapid systematic review, we searched six electronic databases (Medline, Embase, Web of Science, PsycINFO, Cochrane Database of Systematic Reviews and SCOPUS) from inception to April 2020 for systematic reviews appraising interventions for loneliness and/or social isolation. Primary studies from those reviews were eligible if they included: 1) participants in a non-hospital setting; 2) interventions to reduce social isolation and/or loneliness that would be feasible during COVID-19 shielding measures; 3) a relevant control group; and 4) quantitative measures of social isolation, social support or loneliness. At least two authors independently screened studies, extracted data, and assessed risk of bias using the Downs and Black checklist. Study registration: PROSPERO CRD42020178654. We identified 45 RCTs and 13 non-randomised controlled trials; none were conducted during the COVID-19 pandemic. The nature, type, and potential effectiveness of interventions varied greatly. Effective interventions for loneliness include psychological therapies such as mindfulness, lessons on friendship, robotic pets, and social facilitation software. Few interventions improved social isolation. Overall, 37 of 58 studies were of "Fair" quality, as measured by the Downs & Black checklist. The main study limitations identified were the

**Funding:** The author(s) received no specific funding for this work. JG was funded by the National Institute for Health Research Applied Research Collaboration East of England (grant RNAG/564) for time spent on this project. The funder had no role in study design, data collection and analysis, decision to publish, or preparation of the manuscript. The views expressed are those of the authors and not necessarily those of the NIHR or the Department of Health and Social Care.

**Competing interests:** The authors have declared that no competing interests exist.

inclusion of studies of variable quality; the applicability of our findings to the entire population; and the current poor understanding of the types of loneliness and isolation experienced by different groups affected by the COVID-19 pandemic.

## Conclusions

Many effective interventions involved cognitive or educational components, or facilitated communication between peers. These interventions may require minor modifications to align with COVID-19 shielding/social distancing measures. Future high-quality randomised controlled trials conducted under shielding/social distancing constraints are urgently needed.

## Introduction

On 11 March 2020, the World Health Organisation declared the global spread of coronavirus disease 2019 (COVID-19) a pandemic [1]. Countries around the world established escalating containment measures to reduce virus transmission, including travel bans, closure of country borders and lockdowns. In the United Kingdom, over 1.5 million people were told they must self-isolate or "shield" themselves for a period of at least 12 weeks [2]. In addition, strict social distancing guidance both in the UK and internationally advised the public to eliminate all non-essential travel and stay at home [3]. While these measures were initially eased, social distancing measures remain in place, cases and contacts are required to self-isolate, and further national lockdowns have been re-introduced across the world [4–6]. To date, there has been limited literature evaluating the available interventions to protect the mental health of people asked to quarantine, socially distance, or shield during the COVID-19 pandemic. This has prompted a call for high quality research on the effects of COVID-19 on mental health and how to mitigate them [7].

One possible consequence of both the shielding of vulnerable people, and the social distancing restrictions for all, is for physical separation to lead to social isolation and loneliness [8]. Social isolation refers to the objective lack of interaction with others [9]. The concept of loneliness is similar, but refers more generally to the subjective feeling of being alone [10]. Early evidence suggests almost one quarter of adults in the UK have experienced loneliness when living under lockdown [11], while the average person's daily number of contacts has been reduced by up to 74% [12].

There is strong evidence that both social isolation and loneliness are associated with increased all-cause mortality, cardiovascular disease, depression and anxiety [13]. With large numbers worldwide at risk of social isolation and loneliness as a result of the COVID-19 pandemic, there is an urgent need to identify effective interventions to combat this public health problem. Despite the considerable existing literature on interventions that alleviate social isolation or loneliness, many interventions may not be compatible with shielding or social distancing. To provide decision-makers with the evidence needed to tackle this public health challenge, we conducted a rapid systematic review of interventions that treat social isolation and loneliness. We aimed to evaluate the current evidence-base for interventions deemed compatible with shielding/social distancing measures, and to use this to inform public health policy about the most effective types of intervention.

## Methods

### Search strategy and selection criteria

We conducted a rapid systematic review to provide a timely evidence synthesis to urgently inform healthcare policy decisions in the context of the COVID-19 pandemic. We followed established guidelines for conducting rapid systematic reviews [14]. The protocol was registered with the PROSPERO international prospective register of systematic reviews (CRD42020178654; https://www.crd.york.ac.uk/prospero/display_record.php?ID= CRD42020178654) and this review was reported according to the PRISMA statement [15].

We used a two-stage process to identify relevant primary studies. First, we searched Medline, Embase, Web of Science, PsycINFO, Cochrane Database of Systematic Reviews, and SCOPUS databases from inception to April 2020 for relevant systematic reviews. One author (VP) developed and conducted the search with input from CW and JUS. The following search terms were used: ("social isolat*" OR "patient isolat*" OR "emotional isolat*" OR quarantine OR "social distanc*" OR "social support" OR lonel* OR aloneness OR solitude) AND (effect* OR efficien* OR evidence OR consequence* OR impact* OR harm* OR outcome*) AND (intervention* OR promotion* OR program* OR programme* OR campaign* OR prevention) AND ("systematic review*" OR "meta-analys*"). Results in all databases were limited to English language only. Our full search strategy can be found in S1 Appendix in S1 File. Any systematic review reporting interventions that reduce social isolation/loneliness was included. Reference lists of included reviews were screened for additional relevant reviews.

Primary intervention studies from eligible systematic reviews were then retrieved and screened in the second stage, according to the following eligibility criteria. Population: participants of any age in a non-hospital setting; Intervention: all types of intervention to reduce social isolation and/or loneliness that are feasible during COVID-19 shielding measures; Comparison: relevant control group; Outcome: quantitative changes in levels of social isolation, social support or loneliness. The preprint archive MedRxiv was also searched for grey literature relating to isolation, mental health, and COVID-19 using the following search string: (covid-19 or covid19 or coronavirus or corona virus) and (loneliness or coping or mental health) and (isolation). In addition, the titles and abstracts of articles filed in the MedRxiv *COVID-19* and *Psychiatry and Clinical Psychology* subsections were screened for relevant primary studies. Search results were exported to EndNote reference manager and duplicates removed. Rayyan QCRI web app was used to record decisions on included studies [16].

There are many instruments available that assess different aspects of social relationships. We used the framework provided by Valtorta et al to identify and categorise appropriate instruments [17]. We chose three categories to report: 1) measures of loneliness, which include subjective questions on the function of relationships; 2) measures of social isolation or social networks, which use objective, structural measures; and 3) measures of social support, which describe both the function and structure of relationships to varying degrees depending on instrument.

Reviews and primary studies that were solely aimed at patients with specific diseases (e.g Alzheimer's, psychosis) or at minority subgroups of the population (e.g caregivers, divorced parents, bereaved individuals, soldiers, patient relatives) were excluded due to the limited applicability of interventions targeting these groups to the wider public. Only studies with a randomised (including cluster designs) or non-randomised control group were included; pre-post studies without control were excluded.

Each intervention was independently classed by two reviewers (CW and MK) according to alignment with COVID-19 shielding advice. We used March 2020 UK government guidance to inform decisions on feasibility of interventions [18]. In this guidance, shielding is defined as

the avoidance of any face-to-face contact with other people outside one's household. This advice is aimed at people medically defined as extremely vulnerable to COVID-19, whereas the wider public are advised to stay at home if possible and to maintain social distancing of two metres. We chose to use the stricter shielding guidelines to apply feasibility judgements so that interventions would be applicable to the whole population. Interventions originally conducted in a manner not in accordance with COVID-19 shielding guidance, but which may be feasible with minor modifications to the intervention protocol (e.g delivery via videoconferencing), were classed as *Potentially feasible*. Studies of interventions with *Unclear* feasibility were labelled as such, with reasons provided. Interventions were deemed *Not feasible* if physical contact between participants and others is considered an integral part of the intervention.

## Data extraction and synthesis

Two independent reviewers (from CW, AT, MK and RN) double screened titles and abstracts. Where a definite decision to exclude could not be made, full-texts of the systematic reviews were retrieved and screened. Differences were discussed and a consensus reached; a third reviewer was used to resolve disagreements. We (CW, AT and MK) then retrieved and double screened primary studies from each included systematic review to establish whether they met the eligibility criteria.

   Two reviewers extracted data using a pre-designed data extraction sheet to allow standardised reporting of results across studies. We extracted information about: (1) study characteristics including year, location, study design, target participants, age and gender; (2) the intervention; (3) total number of participants in intervention and control groups; (4) intervention duration and follow-up; and (5) study outcomes. Where possible, change-from-baseline effect sizes were calculated using Morris' 2008, *Eq 8* method for estimating effect size from pretest-posttest-control group designs [19]. The direction of effect sizes was standardised so that a positive value indicates improvement. We were unable to perform a meta-analysis due to the heterogeneity of interventions and the incomplete effect size data. Instead, we conducted a narrative synthesis of evidence for interventions affecting the three outcomes described above: loneliness, social isolation and social support.

## Intervention categories

Interventions were categorised using the framework outlined by Gardiner et al [20], which describes six groups using thematic analysis based on the purpose and mechanism of action: social facilitation interventions; psychological therapies; health and social care provision; animal interventions; befriending interventions; and leisure/skill development. The social facilitation category describes interventions with the main purpose of facilitating social interaction between peers, aiming to mutually benefit all involved participants. This contrasts befriending interventions, where the focus is on forming new friendships usually with volunteers to support the lonely individual. Psychological therapies use trained therapists to deliver recognised psychological or cognitive interventions, while health and social care provision involves support from health or social care professionals. Animal interventions use real or artificial animals as the focus of the intervention, while the leisure/skill development category is a broad classification of interventions that provide leisure activities or promote learning a new skill. We used an additional category, educational programme, for interventions that mainly seek to educate participants on topics relevant to social isolation/loneliness, or on health and well-being more generally.

### Risk of bias assessment

Two reviewers (CW and AF) independently assessed risk of bias. We used the Downs and Black tool [21] due to its suitability for both randomised and non-randomised studies. Differences of opinion were resolved by consensus. Downs and Black score ranges were given the following quality levels: excellent (26–28); good (20–25); fair (15–19); and poor ($\leq$14).

## Results

Fig 1 summarises the search and selection process. The systematic literature search retrieved 2914 unique titles/abstracts. We retrieved and screened 159 at full-text level and included 57 relevant systematic reviews. Bibliography searches of these 57 systematic reviews identified a further 10 eligible systematic reviews. From the 67 included systematic reviews, a total of 687 full-text articles were screened and 604 excluded, leaving 83 articles reporting on 81 randomised and non-randomised controlled studies for analysis.

From these 81, twelve studies reported interventions deemed *Feasible* under COVID-19 shielding guidelines. These include videoconferencing programs (n = 2), telephone befriending (n = 2), animal interventions (n = 3), a task framing intervention (n = 1) and several online/virtual programs (n = 4). In 34 studies, interventions were classed as *Potentially Feasible*. For these interventions, it was considered that the core part of the intervention could be conducted remotely using telephone or video call technology. For 12 interventions, feasibility was *Unclear* due to uncertainty over the degree of physical contact required.

A further twenty-three interventions were deemed *Not feasible* or only *Part feasible* with shielding guidelines due to the requirement for physical contact and/or interaction with participants. These include ten health and social care or befriending interventions that typically involved home visits, five leisure/skill development interventions, four animal interventions, three multi-component educational programmes, and a senior centre group programme. Details of these interventions are provided in S1 Table in S1 File, and could potentially be applicable to less stringent physical distancing measures, but are excluded from the analysis below.

Of the 58 included studies, 45 were randomised controlled trials and 13 were non-randomised controlled or quasi-experimental studies. None of the studies had been conducted during the COVID-19 pandemic. Main study characteristics including target participants, setting, age and gender distribution are reported in S2 Table in S1 File. There was considerable heterogeneity in the nature and type of interventions identified. The *Leisure/skill development* category had the greatest number of interventions reported (n = 20), followed by *Psychological therapy* (n = 14), *Educational programmes* (n = 8), *Social facilitation* (n = 7), *Animal interventions* (n = 3), *Befriending interventions* (n = 3), and *Health and Social Care provision* (n = 3).

Quality assessment using the Downs and Black tool revealed many studies (n = 37) were of "Fair" quality (S3 Table in S1 File). 14 studies were judged to be "Good" quality with low risk of bias, while seven were judged to be "Poor" quality studies. Common concerns include a lack of blinding and insufficient reporting of participant loss to follow up: only 11/58 studies reported detailed information on the characteristics of participants lost to follow up, while 32/58 studies did not account for missing follow-up data in their analysis. Due to the nature of interventions, most studies did not blind participants to trial arm, leading to a high risk of performance bias, while detail on blinding of researchers was often missing.

### Intervention effects on loneliness

Loneliness was the most frequently measured outcome, used in 45 studies (Tables 1 and 2). Most studies used established questionnaires when assessing loneliness, including the UCLA

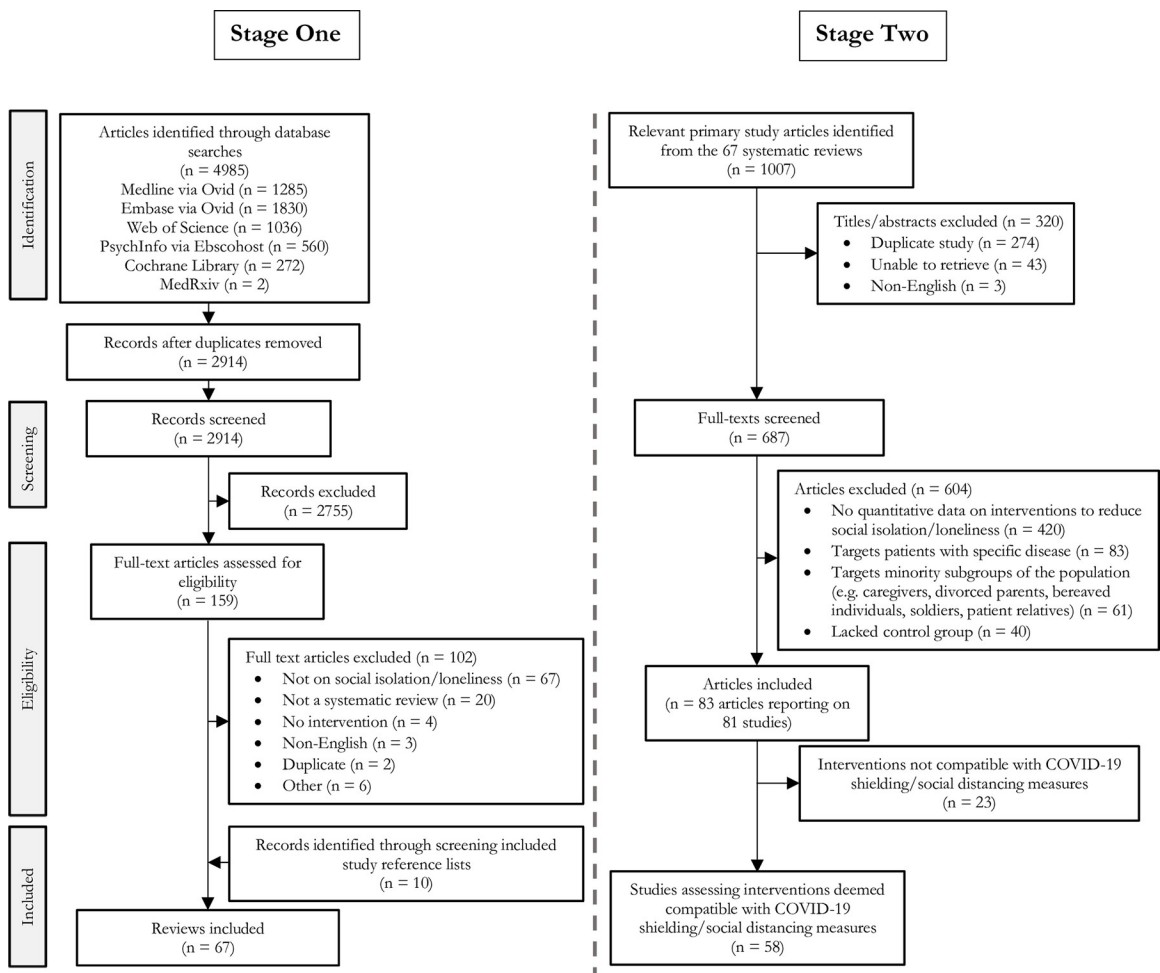

**Fig 1. PRISMA flow diagram.**

Loneliness scale and the De Jong Gierveld Loneliness scale; a minority used generic questions such as "Do you feel lonely?" [22, 23]. Ten studies reported social isolation and/or social support outcome measures in addition to loneliness.

Among the most effective interventions for loneliness were those in the *Psychological therapies* category. Two good quality RCTs of mindfulness-based interventions demonstrated a significant improvement in loneliness [24, 25], as did a weekly Tai Chi Qigong meditation class [26], and a laughter therapy intervention [27]. There were mixed results for reminiscence therapy, where events and experiences from the past are discussed—one RCT demonstrated significant improvement in loneliness scores compared to standard care [28], whereas Westerhof (2018) reported improvement using per-protocol but not intention-to-treat analysis [29]. Two cognitive-based interventions resulted in improved loneliness scores [30, 31], while two others had no significant effect [32, 33].

Most *Animal interventions* were deemed non-feasible, but two robot-based animal studies significantly improved UCLA Loneliness scores [34, 35]. The latter study found that both robotic dogs and living dogs led to similar reductions in loneliness compared to no intervention, but was judged to be of poor quality. One avian companionship intervention involving a live budgerigar did not report significant results [36]. None of the three studies reporting

**Table 1. Summary of results.**

| Intervention category | No. of studies | Studies with significant positive effect | Studies with no significant effect |
|---|---|---|---|
| **Loneliness** | | | |
| **Animal intervention** | 3 | 1 Fair quality RCT [35], 1 Poor quality RCT [34] | 1 Fair quality RCT [36] |
| **Befriending intervention** | 3 | 1 Poor quality RCT [40] | 1 Fair quality RCT [39] and 1 discontinued RCT [37, 38] |
| **Educational programme** | | | |
| *Friendship/Social integration education* | 4 | 2 Fair quality RCTs [44, 45] | 2 Fair quality NRCTs [43, 73, 75] |
| *General wellbeing education* | 3 | None | 1 Good quality RCT [90], 1 Fair quality RCT [74], 1 Fair quality NRCT [76] |
| **Health and Social Care provision** | 2 | None | 1 Good quality RCT [41] 1 Fair quality RCT [42] |
| **Leisure/skill development intervention** | | | |
| *Exercise intervention* | 4 | None | 1 Good quality RCT [58], 3 Fair quality RCTs [56, 57, 59] |
| *Computer training* | 5 | 1 Fair quality NRCT [91] | 3 Fair quality RCTs [53–55], 1 Fair quality NRCT [52] |
| *Video gaming* | 3 | 2 Fair quality RCTs [60, 61] | 1 Fair quality NRCT [62] |
| *Gardening* | 1 | 1 Fair quality RCT [63] | None |
| *General activities* | 1 | None | 1 Fair quality RCT [64] |
| **Psychological therapy** | | | |
| *Mindfulness intervention* | 2 | 2 Good quality RCTs [24, 25] | None |
| *Reminiscence therapy* | 2 | 1 Fair quality RCT [28] | 1 Fair quality RCT [29] |
| *Cognitive-based intervention* | 4 | 1 Fair quality RCT [30], 1 Poor quality NRCT [31] | 1 Good quality RCT [33], 1 Fair quality RCT [32] |
| *Laughter therapy* | 1 | 1 Fair quality NRCT [27] | None |
| *Tai Chi Qigong meditation* | 1 | 1 Good quality RCT [26] | None |
| **Social facilitation** | | | |
| *Videoconference program* | 2 | 1 Good quality RCT [47], 1 Fair quality RCT [48] | None |
| *Group meetings/discussions* | 2 | None | 2 Poor quality RCTs [49, 50] |
| *Other* | 2 | 2 Good quality RCTs [46, 51] | None |
| **Social isolation** | | | |
| **Befriending intervention** | 1 | None | 1 Fair quality RCT [39] |
| **Leisure/skill development intervention** | | | |
| *Exercise intervention* | 3 | None | 1 Good quality RCT [69], 2 Fair quality RCTs [68, 70] |
| *Computer training* | 1 | None | 1 Fair quality RCT [55] |
| *Gardening* | 2 | 1 Fair quality RCT [63] | 1 Poor quality NRCT [66] |
| *General activities* | 1 | 1 Fair quality RCT [64] | None |
| **Psychological therapy** | | | |
| *Logotherapy* | 1 | 1 Fair quality RCT [67] | None |
| *Tai Chi Qigong meditation* | 1 | None | 1 Good quality RCT [26] |
| **Social facilitation** | | | |
| *Mutual help network* | 1 | None | 1 Fair quality NRCT [65] |
| *Group meetings/discussions* | 1 | 1 Poor quality RCT [49] | None |
| *Other* | 2 | None | 2 Good quality RCTs [46, 51] |
| **Social support** | | | |
| **Befriending intervention** | 1 | None | 1 Fair quality RCT [39] |
| **Educational programme** | 1 | None | 1 Fair quality NRCT [92] |
| **Health and Social Care provision** | 1 | None | 1 Fair quality NRCT [93] |
| **Leisure/skill development intervention** | | | |
| *Exercise intervention* | 2 | 1 Good quality RCT [94] | 1 Good quality RCT [69] |

*(Continued)*

**Table 1.** (Continued)

| Intervention category | No. of studies | Studies with significant positive effect | Studies with no significant effect |
|---|---|---|---|
| *Computer training* | 2 | *None* | *1 Fair quality RCT* [55], *1 Fair quality NRCT* [52] |
| *Video gaming* | 1 | *None* | *1 Fair quality NRCT* [95] |
| **Psychological therapy** | | | |
| *Mindfulness intervention* | 1 | *1 Fair quality RCT* [71] | *None* |
| *Cognitive-based intervention* | 2 | *1 Poor quality NRCT* [31] | *1 Good quality RCT* [96] |
| *Tai Chi Qigong Meditation* | 1 | *1 Good quality RCT* [26] | *None* |
| *Visual art discussions* | 1 | *1 Poor quality RCT* [72] | *None* |
| **Social facilitation** | | | |
| *Videoconference program* | 2 | *None* | *1 Good quality RCT* [47], *1 Fair quality RCT* [48] |
| *Mutual help network* | 1 | *None* | *1 Fair quality NRCT* [65] |
| *Other* | 2 | *2 Good quality RCTs* [46, 51] | *None* |

NRCT = non-randomised controlled trial; RCT = randomised controlled trial

*Befriending interventions* showed significant effects: Mountain's (2014) study ended early due to inadequate recruitment [37, 38]; Heller (1991) found no significant improvement in loneliness [39]; and Schulz (1976) reported a significant difference but at a p value of < 0.063 [40]. Additionally, neither of the two *Health and Social Care provision* interventions were shown to reduce loneliness [41, 42].

The content of different *Educational programme* interventions varied—some focused on theories of loneliness and social integration while others sought to educate on health and well-being more generally. Lessons on friendship and social integration typically decreased loneliness, with three of four studies showing improvement in De Jong Gierveld Loneliness scores compared to control [43–45], though the improvement in Tilburg's (2000) study did not reach statistical significance [43].

Conflicting evidence was found in support of *Social facilitation* interventions to reduce loneliness. One high quality randomised controlled study of a dedicated software program (PRISM) featuring internet access, resource guides and an email feature intended to foster connectivity showed significantly decreased loneliness scores post-intervention [46]. Two lower quality cluster-randomised studies demonstrated the effectiveness of videoconferencing programs [47, 48], while two out of three studies of group meetings and/or networking between peers were not found to reduce loneliness [49, 50]. The third of these studies [51] which did report a significant result was substantially higher in quality and involved a group-based educational, cognitive and social support programme designed to improve community knowledge and networking.

Interventions in the *Leisure/skill development* category varied greatly, and many were not effective. Among these are four out of five computer training interventions covering basic computer use, email and internet applications [52–55], and four exercise-related interventions [56–59]. One of these exercise programmes (McAuley 2000) compared aerobic exercise with stretching and toning, and reported improved loneliness in both groups at 6 but not 12 months [56]. Similarly, Dowd et al. (2014) compared two exercise groups which framed exercise either as beneficial for social skills or as beneficial for health—reduced loneliness was found in both exercise groups post-intervention, but with no difference between them [59]. In contrast, two of three gaming interventions were found to be effective at reducing loneliness [60, 61], while the third compared gaming alone and gaming with either an adolescent or elderly person,

**Table 2. Intervention effects on loneliness.**

| Author, year Country | Intervention/Control description | Feasible with COVID-19 shielding/ social distancing? | Loneliness measure | Loneliness result (d/w/m/y = day/week/month/year) |
|---|---|---|---|---|
| **Animal interventions** | | | | |
| Banks, 2008 [34] USA (RCT, n = 38) | I: Animal assisted therapy–weekly visits from either AIBO, a robotic dog (I1), or a living dog (I2) | Yes | UCLAv3 | 7 w: Significant improvement, p < 0.01 |
| | C: Usual care | | | |
| Jessen, 1996 [36] USA (RCT, n = 40) | I: Avian companionship provided by a budgerigar in a cage; participants could interact with the bird but did not have to care for it | Yes | UCLA | 10 d: No significant difference (ES = -0.21) |
| | C: Usual care | | | |
| Robinson, 2013 [35] New Zealand (RCT, n = 40) | I: Weekly sessions with Paro, an advanced interactive robot modelled after a baby Canadian harp seal that responds to contact and other stimuli by moving or imitating the noises of a baby harp seal | Yes | UCLAv3 | 12 w: Significant improvement, p = 0.033 (ES = 0.65) |
| | C: bus trips around the city or alternative activity (crafts, movies, or bingo) | | | |
| **Befriending intervention** | | | | |
| Heller, 1991 [39] USA (RCT, n = 291) | I: Telephone befriending–interviewers called twice a week for 5 weeks and then once a week for 5 weeks, inquiring about the respondent's health and well-being and discussed topics raised by the respondent. After 10 weeks, respondents were randomly assigned to continue staff contact, or to be initiators or recipients of peer telephone contacts (where respondents call each other) | Yes | PLS-7 | 5 w/10 w/20 w/30 w: No significant difference |
| | C: Usual care | | | |
| Mountain, 2014 [37, 38] UK (RCT, n = 70) | I: Telephone befriending–initial one-to-one befriending involved 10 to 20 minute calls weekly aiming to familiarise the participant with the volunteer, conduct everyday conversation and prepare participants for the telephone friendship group. Subsequent friendship groups involved up to 6 participants and 1 hour telephone conferences weekly | Yes | DJGL | Trial ended early (discontinued feasibility trial) |
| | C: Usual care | | | |
| Schulz, 1976 [40] USA (RCT, n = 40) | I: Residents were visited at home by students stating they wished to have first-hand interaction experience with elderly individuals. Participants either controlled visits (I1), were told when they'd be visited (I2), were visited randomly (C1) or were not visited (C2) | Potentially–possible using audio/ video call software | % of time lonely | 2 m: Non-significant improvement, p < 0.063 |
| **Educational programme** | | | | |
| Bouwman, 2017 [44] Netherlands (RCT, n = 239) | I: Online Friendship Enrichment Program– general introduction on friendship and lessons on making new contacts, maintaining relationships, spending time alone, becoming a better friend, and expectations in friendship | Yes | DJGL-SL, DJGL-EL | 6w: DJGL-SL—significant improvement, p < 0.05 (ES = 0.27); DJGL-EL—significant improvement, p < 0.05 (ES = 0.35) |
| | C: Light-full group | | | |

(*Continued*)

**Table 2.** (Continued)

| Author, year Country | Intervention/Control description | Feasible with COVID-19 shielding/ social distancing? | Loneliness measure | Loneliness result |
|---|---|---|---|---|
| | | | | (d/w/m/y = day/week/month/year) |
| Martina, 2006 [73, 75] | I: Friendship enrichment programme–lessons structured according to a 4-stage model on the effect of relational competence in different phases of relationships | Potentially–possible using audio/ video call software | DJGL | 3 m/9 m: No significant difference, p = 0.509 (3 m ES = 0.05; 9 m ES = 0.16) |
| Netherlands | C: Usual care | | | |
| (NRCT, n = 115) | | | | |
| Tilburg, 2000 [43] | I: Friendship programme–lessons focused on different topics related to friendship, consisting of theory on the topic, practice in skills important in friendship, role playing of difficult social situations, and a homework assignment | Unclear–role play element may be difficult via video call | DJGL | 12 m: Non-significant improvement, p = 0.054 (ES = 0.36) |
| Netherlands | | | | |
| (NRCT, n = 64) | | | | |
| | C: Usual care | | | |
| Cohen-Mansfield, 2018 [45] | I: I-SOCIAL intervention addressing psychosocial and environmental barriers to social integration | Potentially–possible using audio/ video call software | UCLA | 6 m/9 m: Significant improvement, p < 0.05 (6 m ES = 0.45; 9 m ES = 0.36) |
| Israel | C: Usual care | | | |
| (RCT, n = 74) | | | | |
| Kremers, 2006 [74] | I: Educational meetings guided by the Self-Management of Well-being (SMW) theory | Potentially–possible using audio/ video call software | DJGL | 6 w/6 m: Non-significant improvement (6 w ES = 0.19; 6 m ES = 0.34) |
| Netherlands | C: Usual care | | | |
| (RCT, n = 142) | | | | |
| Mountain, 2017 [90] | I: Lifestyle Matters–selected topics were explored through discussion, activities and community enactment, with an emphasis on identification of participants' goals, empowerment through sharing strengths/ skills, and providing support to allow participants to practice new or neglected activities independently | Potentially–possible using audio/ video call software | DJGL | 6 m: No significant difference, p = 0.201 (ES = 0.03). 24 m: Significant improvement, p = 0.026 (ES = 0.17) |
| UK | | | | |
| (RCT, n = 288) | | | | |
| | C: Usual care | | | |
| Seepersad, 2005 [76] | I: LUV (Lonely? Unburdening your Vulnerability) programme–psychoeducation with five main activities: program information modules, assignments, weekly discussions, keeping a journal, and an online website for additional resources | Yes/potentially–majority of intervention is feasible without modification; weekly discussions possible using audio/video call software | UCLA | 6 m/9 m: No significant difference, p = 0.08 |
| USA | | | | |
| (NRCT, n = 380) | C: Usual care | | | |
| **Health and Social Care provision** | | | | |
| Hall, 1992 [41] | I: Home visit—Frail Elders Personalised Program–development and review of personal health plan covering healthcare, substance use, exercise, nutrition, stress management, emotional functioning, social support and participation, housing, finances and transportation | Potentially–original protocol required a nurse home visit, but this may be possible using audio/video call software | UCLA | 36 m: No significant difference, p > 0.35 (ES = -0.10) |
| Canada | | | | |
| (RCT, n = 167) | | | | |
| | C: Standard Long Term Care programme | | | |
| van Rossum, 1993 [42] | I: Home visit—visits by nurse to discuss health topics in a broad sense and provide information and advice; participants could also contact the nurse by telephone every day to discuss problems or to ask for an extra visit | Potentially–possible using audio/ video call software | DJGL | 1.5 y/3 y: No significant difference |
| Netherlands | | | | |
| (RCT, n = 580) | C: Usual care | | | |
| **Leisure/skill development intervention** | | | | |

(*Continued*)

**Table 2.** (Continued)

| Author, year Country | Intervention/Control description | Feasible with COVID-19 shielding/ social distancing? | Loneliness measure | Loneliness result (d/w/m/y = day/week/month/year) |
|---|---|---|---|---|
| Shapira, 2007 [91] Israel (NRCT, n = 48) | I: Computer training–educational programme equipped participants with skills for operating a computer and using Internet applications such as email, web browsing, and participating in forums and virtual communities | Potentially–requires computer access. Intervention possible using audio/video call software | UCLA | 17–19 w: Significant improvement, p < 0.001 (ES = 1.58) |
| | C: Activities such as painting, sewing and needlework, and ceramics | | | |
| Slegers, 2008 [54] Netherlands (RCT, n = 236) | I: Computer training–plenary discussions of computer and Internet topics (basic computer use, Internet applications e.g email, search engine), followed by individual assignments from a workbook | Yes/Potentially–requires computer access. Intervention possible using audio/video call software | DJGL | 4 m/12 m: No significant difference, p = 0.84 (4 m ES = -0.17; 12 m ES = -0.09) |
| | C1: No training and no intervention (usual care), C2: Non-interested participant control group (usual care) | | | |
| White, 1999 [52] USA (NRCT, n = 27) | I: Computer training–basic training in computer use, an introduction to the use of email and the Internet, and basic instruction in word processing | Potentially–requires computer access. Intervention possible using audio/video call software | UCLAv3 | 2 w: Significant improvement, p = 0.04; 5 m: No significant difference, p = 0.17 (ES = 0.52) |
| | C: Usual care | | | |
| White, 2002 [53] USA (RCT, n = 100) | I: Computer training–covering basic computer operation, use of email, and an introduction to accessing the World Wide Web | Potentially–requires computer access. Intervention possible using audio/video call software | UCLAv3 | 5 m: No significant difference, p = 0.52 |
| | C: Usual care | | | |
| Woodward, 2011 [55] USA (RCT, n = 82) | I: Computer training–tutorial sessions on topics ranging from the basics of computer use to blogging, manipulating photos, and using voice/video via the Internet | Potentially–requires computer access. Intervention possible using audio/video call software | DJGL | 3 m/6 m/9 m: No significant difference |
| | C: Usual care | | | |
| Bickmore, 2005 [57] USA (RCT, n = 21) | I: Virtual exercise advisor ("Relational agent")—an animated character that simulates face-to-face conversation with users, playing the role of an exercise advisor that interacts with users on a daily basis to motivate them to exercise more through walking | Yes | UCLA | 2 m: No significant difference, p = 0.12 |
| | C: Usual care | | | |
| Dowd, 2014 [59] Canada (RCT, n = 84) | I: Framing exercise as beneficial for social skills–participants asked to read a physical activity guide and given a task framing information sheet detailing that engaging in regular exercise is indicative of self-regulation, which is associated with characteristics that are important for social relationships | Yes | UCLAv3 | 4 w: No significant difference between different framing groups, p ≥ 0.21; significant improvement in loneliness in both groups |
| | C: Framing physical exercise as beneficial for health | | | |
| McAuley, 2000 [56] USA (RCT, n = 174) | I1: Exercise programme–aerobic exercise classes that employed brisk walking as the aerobic component | Unclear–group walking requires physical contact | UCLA | 6 m/12 m: No significant difference between groups; both groups showed significantly improved loneliness immediately post intervention (6 m) which returned to baseline levels at 12 m |
| | I2: Exercise programme–stretch and toning classes focused on stretching, limbering and mild strengthening for the whole body | | | |

*(Continued)*

**Table 2.** (Continued)

| Author, year Country | Intervention/Control description | Feasible with COVID-19 shielding/ social distancing? | Loneliness measure | Loneliness result (d/w/m/y = day/week/month/year) |
|---|---|---|---|---|
| Mutrie, 2012 [58] UK (RCT, n = 41) | I: Exercise programme–Individualised walking programme in the form of a specially designed booklet and pedometer. A walking group that met twice weekly was also an option for participants | Unclear–group walking requires physical contact | UCLA | 12 w/24 w: No significant difference |
| | C: Usual care | | | |
| Tse, 2010 [63] Hong Kong (RCT, n = 53) | I: Indoor gardening programme–participants were provided with equipment and given their own plants to look after, taught how to make natural pesticides using readily accessible raw materials, took photos and shared planting diaries | Potentially–possible using audio/ video call software | UCLA | 8 w: Significant improvement, p = 0.00 (ES = 0.94) |
| | C: Usual care | | | |
| Kahlbaugh, 2011 [61] USA (RCT, n = 35) | I: Gaming–participants played a Wii game of their choice (everyone chose Wii bowling) for 1 hour a week | Unclear–protocol involved group play. Whether similar findings can be extended to online play is unclear | UCLAv3 | 10 w: Significant improvement, p < 0.05 |
| | C1: Watching TV with a partner; C2: No visit control | | | |
| Jung, 2009 [60] Singapore (RCT, n = 45) | I: Gaming–participants were given a Wii set with controllers and played four games from Wii Sports and Cooking Mama | Unclear–protocol involved group play. Whether similar findings can be extended to online play is unclear | UCLAv3 | 6 w: Significant improvement, p < 0.01 (ES = 1.00) |
| | C: Traditional games (memory games, UNO, Jenga) | | | |
| Xu, 2016 [62] Singapore (NRCT, n = 89) | I: Gaming–participants were asked to play three Kinect exergames, with each exergame being played for 10 to 15 minutes. Exergames only required simple and repetitive action so were suitable for older adults | Unclear–protocol involved group play. Whether similar findings can be extended to online play is unclear | UCLAv3 | 1 w: No significant difference between playing alone vs with an elderly person or adolescent, p = 0.878 |
| | C: playing alone | | | |
| Winstead, 2014 [64] USA (RCT, n = 141) | I: Twice weekly 90 minute sessions participating in informal activities in assisted and independent living communities. Study sought to determine the effect of activities of any form on outcomes, not to examine the type of activity | Unclear–activity descriptions not provided | UCLA-3 | 3 m: No significant difference (ES = 0.13) |
| | C: Usual care | | | |
| **Psychological therapy** | | | | |
| Conoley, 1985 [32] USA (RCT, n = 57) | I: Cognitive reframing– 2 sessions, one week apart, aiming to change the way participants view their experiences | Potentially–possible using audio/ video call software | UCLA | 1 w/3 w: No significant difference (1 w ES = 0.97; 3 w ES = 0.57) |
| | C: Usual care | | | |
| Dodge, 2015 [33] USA (RCT, n = 83) | I: Web-enabled conversational interactions to improve cognitive functions–face-to-face conversations with trained interviewers 5 days a week for 6 weeks by way of a dedicated video-chat-enabled PC provided to each subject | Yes | HLS-3 | 8 w/18 w: No significant difference, p = 0.44 |
| | C: Usual care | | | |

(*Continued*)

**Table 2.** (Continued)

| Author, year Country | Intervention/Control description | Feasible with COVID-19 shielding/ social distancing? | Loneliness measure | Loneliness result (d/w/m/y = day/week/month/year) |
|---|---|---|---|---|
| McWhirter, 1996 [30] USA (RCT, n = 44) | I: Cognitive behavioural intervention–weekly group experiences involving role play and homework activities. The intimate condition used cognitive and behavioural techniques focused on establishing and maintaining intimate relationships. The social condition combined cognitive restructuring to modify attributional styles and develop better communication skills in social settings | Unclear–role play element may be difficult via video call | UCLA-I, UCLA-S, ILS, SLS | 6 w/14 w: Intimate group–no significant difference. Social group–Significant improvement in UCL-I (p = 0.01), UCLA-S (p = 0.03), ILS (p = 0.004) and SLS (p = 0.04) |
| | C: High-demand control–alternative group experience where feelings and experiences were expressed, but information or activities relevant to the reduction of loneliness was not offered | | | |
| Winningham, 2007 [31] USA | I: Cognitive enhancement programme–sessions designed to educate participants about the brain and memory, stimulate memory, and encourage participants to learn and memorise interesting information about each other | Potentially–possible using audio/ video call software | UCLAv3 | 3 m: Significant improvement (ES = 0.52) |
| (NRCT, n = 58) | C: Usual care | | | |
| Kuru Alici, 2018 [27] | I: Laughter therapy–laughter exercises, deep breathing exercises, playing games, singing songs loudly, laughter meditation | Potentially–possible using audio/ video call software | DJGL | 5 w: Significant improvement, p = 0.000 (ES = 3.05) |
| Turkey | C: Usual care | | | |
| (NRCT, n = 72) | | | | |
| Creswell, 2012 [24] USA (RCT, n = 40) | I: Mindfulness-Based Stress Reduction (MSBR) programme–meditation exercises, mindful yoga and stretching, and group discussions with the intent to foster mindful awareness of one's moment-to-moment experience | Potentially–possible using audio/ video call software | UCLA | 8 w: Significant improvement, p = 0.02, $\eta^2$ = 0.17[†] |
| | C: Usual care | | | |
| Zhang, 2018 [25] China (RCT, n = 50) | I: Mindfulness-based cognitive therapy (MBCT)–participants learned theories, practiced mindfulness exercises, and discussed home practice | Potentially–possible using audio/ video call software | CCSL | 8 w: Significant improvement, p = 0.03, Cohen's D = 0.66[†] (ES = 0.98) |
| | C: Usual care | | | |
| Chiang, 2010 [28] Taiwan (RCT, n = 110) | I: Reminiscence therapy–structured weekly sessions concentrated on a different topic each week, including sharing memories and greeting each other, increasing participant awareness/expression of their feelings, identifying past positive relationships, recalling family history and life stories, and identifying positive strengths and goals | Potentially–possible using audio/ video call software | UCLAv3 | 2 m/5 m: Significant improvement, p < 0.0001 (2 m ES = 0.96, 5 m ES = 0.94) |
| | C: Usual care | | | |
| Westerhof, 2018 [29] Netherlands (RCT, n = 81) | I: Reminiscence intervention–weekly sessions to elicit specific positive memories in childhood, adolescence, and adulthood | Potentially–possible using audio/ video call software | DJGL | 2 m/8 m: No significant difference (2 m and 8 m ES = 0.05) |
| | C: Participants were visited 5 times, engaging in conversation, playing cards, going shopping etc | | | |

(Continued)

 

**Table 2.** (Continued)

| Author, year Country | Intervention/Control description | Feasible with COVID-19 shielding/ social distancing? | Loneliness measure | Loneliness result (d/w/m/y = day/week/month/year) |
|---|---|---|---|---|
| Chan, 2017 [26] Hong Kong (RCT, n = 48) | I: 18 forms of Tai chi qigong–twice weekly tai chi class led by an experienced tai chi qigong instructor whose motions, postures and speed of movement participants had to copy. Participants were also encouraged to self-practice for 30 minutes a day<br><br>C: Usual care | Potentially–possible using audio/ video call software | DJGL | 3 m: No significant difference (ES = 0.25). 6 m: Significant improvement (ES = 0.70) |
| **Social facilitation** | | | | |
| Andersson, 1985 [49] Sweden (RCT, n = 108) | I: Group meetings between participants in the same neighbourhood, discussing the residential area, the role of retiree, social and medical services, and opportunities for leisure activities<br><br>C: Usual care | Potentially–possible using audio/ video call software | UCLA | 6 m: No significant difference, p = 0.073 |
| Czaja, 2018 [46] USA (RCT, n = 300) | I: Personal Reminder Information and Social Management (PRISM) computer software–Internet access, annotated resource guide, classroom, calendar, and photo features, email, games, and online help<br><br>C: Binder group—participants received a binder that contained content similar to that found on PRISM | Yes/Potentially–requires computer access. Protocol involved three home visits for training, but this may be feasible remotely | UCLA | 6 m: Significant improvement; Cohen's d = 0.17 (0.16–3.28)[†], p < 0.04 |
| Lokk, 1990 [50] Sweden (RCT, n = 65) | I: Group discussion on participants' goals and achievement of those goals, with feedback given by participating group members<br><br>C: Usual care | Potentially–possible using audio/ video call software | "Do you feel lonely?" | 6 w: Significant improvement, p = 0.03. 12 w/24 w: No significant difference |
| Saito, 2012 [51] Japan (RCT, n = 60) | I: Educational cognitive and social support programme designed to improve community knowledge and networking with other participants and community gatekeepers<br><br>C: Usual care | Potentially–possible using audio/ video call software | AOKL | 3 m/8 m: Significant improvement, p = 0.011 (3 m ES = 0.56; 8 m ES = 1.46) |
| Tsai, 2010 [48] Taiwan (RCT, n = 57) | I: Videoconference program–weekly videoconference call with main family contact person for 3 months<br><br>C: Usual care | Yes | UCLA | 1 w/3 m: Significant improvement, p = 0.03 (1 w ES = 0.13; 3 m ES = 0.33) |
| Tsai, 2011 [47] Taiwan (RCT, n = 90) | I: Videoconference program–weekly videoconference call with main family contact person for 3 months using laptops, followed by use of program on request after 3 months<br><br>C: Usual care | Yes | UCLA | 3 m/6 m/9 m: Significant improvement, p < 0.001 (3 m ES = 0.50; 6 m ES = 0.55; 9 m ES = 0.63) |

Ordered by intervention type then author. AOKL = Ando–Osada–Kodama loneliness scale; CCSL = Chinese college student loneliness scale; DJGL = de Jong Gierveld loneliness scale; DJGL-EL = de Jong Gierveld loneliness scale, emotional loneliness subscale; DJGL-SL = de Jong Gierveld loneliness scale, social loneliness subscale; ES = effect size (standardised mean difference); HLS-3 = Hughes 3-item Loneliness scale; HSSBS = Hsuing Social Support Behaviours scale; ILS = Intimate Loneliness scale; NRCT = non-randomised controlled trial; PLS-7 = Paloutzian 7-item loneliness scale; RCT = randomised controlled trial; SLS = Social Loneliness scale; SSBS = Social Support Behaviours scale; UCLA = University California Los Angeles loneliness scale (1980); UCLA-3 = University California Los Angeles loneliness scale (3-item); UCLA-4 = University California Los Angeles loneliness scale (4-item); UCLA-I = University California Los Angeles loneliness scale, intimate subscale; UCLA-S = University California Los Angeles loneliness scale, social subscale; UCLAv3 = University California Los Angeles loneliness scale version 3 (1996). [†]Effect size reported in study results.

finding no difference between groups [62]. One study of an indoor gardening programme in a nursing home, where participants were given their own plants and taught how to look after them, reported decreased loneliness scores among participants of the programme [63].

## Intervention effects on social isolation

Fourteen studies reported on social isolation using a variety of instruments that measure isolation, social networks, or number of social contacts (Tables 1 and 3). Most interventions fell under the *Leisure/skill development*, *Psychological* and *Social facilitation* categories, and few reduced social isolation. Notably, a twice weekly activity session decreased social isolation [64], while group meetings between neighbours led to increased social contact despite not significantly altering loneliness levels [49]. In contrast, a mutual help network of residents in an apartment building was not found to significantly increase social ties [65].

Of the two gardening-related interventions, the indoor gardening programme increased participants' social networks within a nursing home [63], whereas a poor quality study evaluating horticultural therapy was not found to improve social connectedness [66]. Logotherapy, a meaning-oriented therapy that helps individuals appreciate their existence, was associated with decreased social disconnectedness and isolation [67], while Tai Chi Qigong classes and Saito's (2012) social support programme did not increase social networks despite improving feelings of loneliness [26, 51]. As previously seen with loneliness outcomes, telephone befriending [39], computer training [55], and exercise programmes had no significant effect on measures of social isolation [68–70].

## Intervention effects on social support

Eighteen studies reported on social support using the Duke Social Support, Perceived Social Support, Multidimensional Perceived Social Support, and Medical Outcomes Study Social Support scales, among others (Tables 1 and 4). Just as for loneliness, *Psychological* interventions were the most successful at increasing social support. In particular, mindfulness therapy [71], visual art discussions [72], Tai Chi Qigong meditation [26], and a cognitive enhancement programme were found to improve social support [31]. In contrast, *Befriending*, *Educational*, and *Health and Social Care provision* interventions did not have any significant effects. Mixed evidence was found for *Social facilitation* interventions that improved social support. Three studies reported significant results—these include the PRISM software program and the social support programme described previously [46, 51], while one of the two videoconferencing programs reported significantly improved social support scores at 1 week but not 3 months [48].

## Effective interventions for specific population groups

Of the 58 included studies, a majority (n = 51) targeted older adults. These studies were typically conducted either in the community, at day-care centres, in nursing homes, or within other types of residential care facility. In total, 17 studies were conducted in nursing or care facilities. Effective interventions in this setting include weekly visits from an interactive robotic dog or seal [34, 35], Wii gaming [60], gardening [63], videoconferencing [47, 48], and cognitive/psychological interventions [27, 28, 31]. A further six interventions were conducted in retirement homes or communities, among which only Wii gaming was found to be effective [61].

There was a female majority among study participants in all but five studies, and seven were exclusively open to female participants [32, 39, 43, 49, 72–74]. Of these, visual art discussions and neighbourhood group meetings were effective at reducing loneliness and social isolation respectively [49, 72], while educational well-being meetings were associated with a non-significant improvement in loneliness [74]. In contrast, conflicting evidence for a friendship enrichment programme was found [43, 73, 75], and a telephone befriending study of female residents in low-income housing was not effective [39].

**Table 3. Intervention effects on social isolation.**

| Author, year, Country | Intervention/Control description | Feasible with COVID-19 shielding/ social distancing? | Social isolation/ network measure | Social isolation/network result (d/w/m/y = day/week/month/ year) |
|---|---|---|---|---|
| **Befriending intervention** | | | | |
| Heller, 1991 [39] USA (RCT, n = 291) | I: Telephone befriending–interviewers called twice a week for 5 weeks and then once a week for 5 weeks, inquiring about the respondent's health and well-being and discussed topics raised by the respondent. After 10 weeks, respondents were randomly assigned to continue staff contact, or to be initiators or recipients of peer telephone contacts (where respondents call each other) C: Usual care | Yes | NE | 5 w/10 w/20 w/30 w: No significant difference |
| **Leisure/skill development intervention** | | | | |
| Woodward, 2011 [55] USA (RCT, n = 82) | I: Computer training–tutorial sessions on topics ranging from the basics of computer use to blogging, manipulating photos, and using voice/video via the Internet C: Usual care | Potentially–requires computer access. Intervention possible using audio/video call software | # in social network, frequency of contact | 3 m/6 m/9 m: No significant difference |
| Kamegaya, 2014 [70] Japan (RCT, n = 52) | I: Exercise and leisure activity programme–physical activity was the primary content, involving muscle-stretching, muscle-strengthening, and aerobic exercise at home. Leisure activities, such as cooking, handcrafts and competitive games, were also included in the programme C: Usual care | Unclear–physical contact may be required | LSNS | 12 w: No significant difference, p = 0.185 (ES = 0.38) |
| Iliffe, 2014 [69] UK (RCT, n = 953) | I1: Otago Exercise Programme (OEP): 30 min programme of leg muscle strengthening and balance retraining exercises and a walking plan. I2: Falls Management Exercise (FaME) programme: 1 hour group exercise class in a local community centre and two 30 min home exercise sessions (based on the OEP) C: Usual care | Potentially (OEP)–protocol involved home visits for training, but this may be feasible remotely. No (FaME)–physical contact required | LSNS | 12 m: No significant improvement (OEP or FaME) (ES [OEP] = -0.09; ES [FaME] = -0.22 |
| Maki, 2012 [68] Japan (RCT, n = 150) | I: Exercise programme—30 minute daily walking and 60 minute group walking excursions C: Educational lectures on food, nutrition, and oral care | Unclear–group walking requires physical contact | LSNS | 3 m: No significant difference, p = 0.16 (ES = 0.21) |
| Perkins, 2012 [66] USA (NRCT, n = 34) | I: Horticultural therapy programme–weekly group classes consisting of 1) Herb of the Day, 2) Learning a planting technique, 3) Main activity (herb-related), and 4) Cooking a snack (using the grown herbs) C: Usual care | Potentially–possible using audio/video call software | HFS | 6 w: No significant difference, p = 0.48 (ES = -0.08) |
| Tse, 2010 [63] Hong Kong (RCT, n = 53) | I: Indoor gardening programme–participants were provided with equipment and given their own plants to look after, taught how to make natural pesticides using readily accessible raw materials, took photos and shared planting diaries C: Usual care | Potentially–possible using audio/video call software | LSNS | 8 w: Significant improvement, p = 0.00 (ES = 0.85) |
| Winstead, 2014 [64] USA (RCT, n = 141) | I: Twice weekly 90 minute sessions participating in informal activities in assisted and independent living communities. Study sought to determine the effect of activities of any form on outcomes, not to examine the type of activity C: Usual care | Unclear–activity descriptions not provided | SIS | 3 m: Significant improvement, p < 0.01 (ES = 0.45) |

*(Continued)*

**Table 3.** (Continued)

| Author, year, Country | Intervention/Control description | Feasible with COVID-19 shielding/ social distancing? | Social isolation/ network measure | Social isolation/network result (d/w/m/y = day/week/month/ year) |
|---|---|---|---|---|
| **Psychological therapy** | | | | |
| Elsherbiny, 2018 [67] Egypt (RCT, n = 43) | I: Logotherapy–a meaning-oriented therapy that aims to help individuals appreciate their existence. The positive consequences and importance of social networks and interactions with others are highlighted | Potentially–possible using audio/video call software | SD; PIS | 12 w/14 w: Significant improvement |
| | C: Usual care | | | |
| Chan, 2017 [26] Hong Kong (RCT, n = 48) | I: 18 forms of Tai chi qigong–twice weekly tai chi class led by an experienced tai chi qigong instructor whose motions, postures and speed of movement participants had to copy. Participants were also encouraged to self-practice for 30 minutes a day | Potentially–possible using audio/video call software | LSNS | 3 m/6 m: No significant difference (3 m ES = 0.05; 6 m ES = 0.38) |
| | C: Usual care | | | |
| **Social facilitation** | | | | |
| Andersson, 1985 [49] Sweden (RCT, n = 108) | I: Group meetings between participants in the same neighbourhood, discussing the residential area, the role of retiree, social and medical services, and opportunities for leisure activities | Potentially–possible using audio/video call software | SCm | 6 m Significant improvement, p = 0.028 |
| | C: Usual care | | | |
| Baumgarten, 1988 [65] Canada (NRCT, n = 95) | I: Mutual help network–a) people willing to volunteer their help were matched with people who required help, b) participants volunteered to plan and coordinate group activities | Unclear–dependant on help required and activities planned | NST | 16 m: No significant difference, p = 0.87 |
| | C: Usual care | | | |
| Czaja, 2018 [46] USA (RCT, n = 300) | I: Personal Reminder Information and Social Management (PRISM) computer software–Internet access, annotated resource guide, classroom, calendar, and photo features, email, games, and online help | Yes/Potentially–requires computer access. Protocol involved three home visits for training, but this may be feasible remotely | HFS | 6 m: Non-significant improvement; Cohen's d = 0.17 (-1.47 to 0.16)[†], p < 0.11 |
| | C: Binder group—participants received a binder that contained content similar to that found on PRISM | | | |
| Saito, 2012 [51] Japan (RCT, n = 60) | I: Educational cognitive and social support programme designed to improve community knowledge and networking with other participants and community gatekeepers | Potentially–possible using audio/video call software | SNm | 3 m/8 m: No significant difference |
| | C: Usual care | | | |

Ordered by intervention type then author. ES = Effect size (standardised mean difference); HFS = Hawthorne Friendship scale (social connectedness); LSNS = Lubben Social Network Scale; NE = Network embeddedness scale; NRCT = Non-randomised controlled trial; NST = Number of social ties; PIS = Perceived isolation scale; RCT = Randomised controlled trial; SCm = Social Contacts measure; SD = Social disconnectedness scale; SIS = Social isolation scale; SNm = Social network measure.
[†]Effect size reported in study results.

Finally, six studies targeted student populations studying at university or college and all involved a psychological or cognitive component. Among the effective interventions were two Mindfulness-based therapies, one trialled in a university community and the other recruiting lonely college students, in addition to a cognitive behavioural intervention at a university counselling centre [25, 30, 71]. Cognitive reframing sessions for female undergraduate psychology volunteers and the "Lonely? Unburdening your Vulnerability" (LUV) programme for college students were ineffective, as were attempts to frame exercise as beneficial for social skills among inactive university students [32, 59, 76].

**Table 4. Intervention effects on social support.**

| Author, year, Country | Intervention/Control description | Feasible with COVID-19 shielding/social distancing? | Social support measure | Social support result (d/w/m/y = day/week/month/year) |
|---|---|---|---|---|
| **Befriending intervention** | | | | |
| Heller, 1991 [39]<br>USA<br>(RCT, n = 291) | I: Telephone befriending–interviewers called twice a week for 5 weeks and then once a week for 5 weeks, inquiring about the respondent's health and well-being and discussed topics raised by the respondent. After 10 weeks, respondents were randomly assigned to continue staff contact, or to be initiators or recipients of peer telephone contacts (where respondents call each other)<br>C: Usual care | Yes | PSSS | 5 w/10 w/20 w/30 w: No significant difference |
| **Educational programme** | | | | |
| Ruffing-Rahal, 1994 [92]<br>USA<br>(NRCT, n = 28) | I: Health promotion–weekly group discussion, health education, and group exercise/wellness<br>C: Usual care | Potentially–possible using audio/video call software | ISSI | 6 m: No significant difference, p = 0.6580 (ES = -0.07) |
| **Health and Social Care provision** | | | | |
| Dickens, 2011 [93]<br>UK<br>(NRCT, n = 393) | I: Devon Community Mentoring Service–participants assigned a mentor who worked closely with clients to build self-confidence and engage in personally meaningful social activities<br>C: Usual care | Potentially–possible using audio/video call software | MOS-SSS | 6 m: No significant difference, p = 0.75 (ES = 0.09) |
| **Leisure/skill development intervention** | | | | |
| White, 1999 [52]<br>USA<br>(NRCT, n = 27) | I: Computer training–basic training in computer use, an introduction to the use of email and the Internet, and basic instruction in word processing<br>C: Usual care | Potentially–requires computer access. Intervention possible using audio/video call software | DSSI | 2 w/5 m: No significant difference, p = 0.32 (ES = -0.32) |
| Woodward, 2011 [55]<br>USA<br>(RCT, n = 82) | I: Computer training–tutorial sessions on topics ranging from the basics of computer use to blogging, manipulating photos, and using voice/video via the Internet<br>C: Usual care | Potentially–requires computer access. Intervention possible using audio/video call software | MSPSS | 3 m/6 m/9 m: No significant difference |
| Iliffe, 2014 [69]<br>UK<br>(RCT, n = 953) | I1: Otago Exercise Programme (OEP): 30 min programme of leg muscle strengthening and balance retraining exercises and a walking plan. I2: Falls Management Exercise (FaME) programme: 1 hour group exercise class in a local community centre and two 30 min home exercise sessions (based on the OEP)<br>C: Usual care | Potentially (OEP)–protocol involved home visits for training, but this may be feasible remotely. No (FaME)–physical contact required | MSPSS | 12 m: No significant improvement (OEP or FaME) (ES [OEP] = -0.27; ES [FaME] = -0.24 |
| Tarazona-Santabalbina, 2016 [94]<br>Spain<br>(RCT, n = 100) | I: Exercise programme– 65 minutes of daily activities 5 days a week involving proprioception/balance exercises, aerobic training, strength training and stretching<br>C: Usual care | Potentially–possible using audio/video call software | DSSI | 6 m: Significant improvement, p < 0.001 (ES = 0.93) |
| Bell, 2011 [95]<br>USA<br>(NRCT, n = 22) | I: Gaming–Nintendo Wii bowling +/- falls education<br>C: Usual care | Unclear–protocol involved group play. Whether similar findings can be extended to online play is unclear | SPS | 8 w: No significant difference in total score; significant improvement in Item #1, 3, 14 |
| **Psychological therapy** | | | | |

*(Continued)*

**Table 4.** (*Continued*)

| Author, year, Country | Intervention/Control description | Feasible with COVID-19 shielding/social distancing? | Social support measure | Social support result (d/w/m/y = day/week/month/year) |
|---|---|---|---|---|
| Cobb, 2014 [96] USA (RCT, n = 1502) | I: "Daily challenge"—a freely accessible, email, web and mobile intervention where participants receive a daily email/test suggesting a small health-related action they can complete in a few minutes, along with information on how to complete the challenge | Yes | ISEL | 1 m/3 m: No significant difference, $p > 0.05$ (1 m ES = 0.02; 3 m ES = 0.08) |
| | C: weekly generic health newsletter | | | |
| Winningham, 2007 [31] USA (NRCT, n = 58) | I: Cognitive enhancement programme–sessions designed to educate participants about the brain and memory, stimulate memory, and encourage participants to learn and memorise interesting information about each other | Potentially–possible using audio/video call software | SS-A/SS-B | 3 m: Significant improvement, $p = 0.001$ (ES = 0.78) (SS-A), $p = 0.02$ (ES = 0.70) (SS-B) |
| | C: Usual care | | | |
| Adair, 2018 [71] USA (RCT, n = 94) | I: Mindfulness Meditation training course–instruction on how to meditate and be mindful in the context of 6 topics: mindfulness of breath, body sensation emotions, thoughts, attitude, and choiceless awareness | Potentially–possible using audio/video call software | Social connection (UCLA, MOS-SSS) | Significant improvement, $p < 0.05$ |
| | C: Health Promotion active control course | | | |
| Chan, 2017 [26] Hong Kong (RCT, n = 48) | I: 18 forms of Tai chi qigong–twice weekly tai chi class led by an experienced tai chi qigong instructor whose motions, postures and speed of movement participants had to copy. Participants were also encouraged to self-practice for 30 minutes a day | Potentially–possible using audio/video call software | RSSQ-NP, RSSQ-S | 3 m: No significant difference. 6 m: Significant improvement in RSSQ-S (ES = 0.59); No significant difference in RSSQ-NP (ES = 0.50) |
| | C: Usual care | | | |
| Wikstrom, 2002 [72] Sweden (RCT, n = 40) | I: Visual art discussions–participants were asked to describe the painting, to use their imagination to describe why, how and when it was made, and to describe associations that appear when looking at the painting such as feelings, memories and thoughts | Potentially–possible using audio/video call software | SICA | Post-intervention/4 m: Significant improvement, $p = 0.0001$ |
| | C: Control group discussions | | | |
| **Social facilitation** | | | | |
| Baumgarten, 1988 [65] Canada (NRCT, n = 95) | I: Mutual help network–a) people willing to volunteer their help were matched with people who required help, b) participants volunteered to plan and coordinate group activities | Unclear–dependant on help required and activities planned | SSS | 16 m: No significant difference, $p = 0.37$ |
| | C: Usual care | | | |
| Czaja, 2018 [46] USA (RCT, n = 300) | I: Personal Reminder Information and Social Management (PRISM) computer software–Internet access, annotated resource guide, classroom, calendar, and photo features, email, games, and online help | Yes/Potentially–requires computer access. Protocol involved three home visits for training, but this may be feasible remotely | ISEL | 6 m: Significant improvement; Cohen's d = 0.28 (-3.26 to -0.66)[†], $p < 0.004$ |
| | C: Binder group—participants received a binder that contained content similar to that found on PRISM | | | |
| Saito, 2012 [51] Japan (RCT, n = 60) | I: Educational cognitive and social support programme designed to improve community knowledge and networking with other participants and community gatekeepers | Potentially–possible using audio/video call software | SSm | 3 m/8 m: Significant improvement, $p = 0.013$ (8 m ES = 0.83) |
| | C: Usual care | | | |

(*Continued*)

**Table 4.** (Continued)

| Author, year, Country | Intervention/Control description | Feasible with COVID-19 shielding/social distancing? | Social support measure | Social support result (d/w/m/y = day/week/month/year) |
|---|---|---|---|---|
| Tsai, 2010 [48] Taiwan | I: Videoconference program–weekly videoconference call with main family contact person for 3 months | Yes | HSSBS | 1 w: Significant improvement, p < 0.01 (ES = 0.27); 3 m: No significant difference, p = 0.23 (ES = 0.21) |
| (RCT, n = 57) | C: Usual care | | | |
| Tsai, 2011 [47] Taiwan | I: Videoconference program–weekly videoconference call with main family contact person for 3 months using laptops, followed by use of program on request after 3 months | Yes | HSSBS | 3 m/6 m/9 m: No significant improvement (3 m ES = 0.21; 6 m ES = 0.03; 9 m ES = 0.20) |
| (RCT, n = 90) | C: Usual care | | | |

Ordered by intervention type then author. DSSI = Duke Social Support Index; ES = Effect size (standardised mean difference); HSSBS = Hsuing Social Support Behaviours scale; ISEL = Interpersonal Support Evaluation List; ISSI = Interaction Schedule for Social Interaction; MOS-SSS = Medical Outcomes Study Social Support Survey; MSPSS = Multidimensional Scale of Perceived Social Support; NRCT = Non-randomised controlled trial; PSSS = Perceived Social Support scale; RCT = Randomised controlled trial; RSSQ-NP = Revised Social Support Questionnaire, total number of people; RSSQ-S = Revised Social Support Questionnaire, total satisfaction; Social connection (UCLA/MOS-SSS) = Social Connection factor incorporating UCLA loneliness scale and MOS-SSS; SICA = Social Interaction Complete Amount; SPS = Social Provisions Scale; SS-A = Social Support Appraisal scale; SS-B = Social Support Behaviours scale; SSBS = Social Support Behaviours scale; SSm = Social support measure; SSS = Social Support satisfaction. [†]Effect size reported in study results

## Discussion

To our knowledge, this is the first systematic review of interventions for social isolation and loneliness that can be applied during the COVID-19 pandemic or other situations where social distancing is required. We identified 58 studies of interventions to reduce social isolation, social support and loneliness that may be feasible with shielding/social distancing guidelines. There was significant heterogeneity in the interventions identified, and we found mixed results across the intervention categories.

Many *Psychological therapy* interventions were effective, with studies of mindfulness-based therapies, Tai Chi Qigong meditation, laughter therapy and visual art discussions demonstrating significant improvements in loneliness or social support outcomes. These represent potentially low-cost interventions that can be conducted in online groups on a large scale. Additionally, while *Educational programme* interventions varied greatly in both procedure and overall results, several studies found that lessons on making friends and addressing barriers to social integration had a positive effect on loneliness. These findings collectively suggest a possible underlying cognitive aspect to loneliness, which may be targeted either directly using psychology-based interventions, or indirectly by exploring the causes of one's loneliness and practising the development and maintenance of social relationships [77].

When considering interventions aiming to increase contact with others, more evidence was found in support of *Social facilitation* interventions compared with *Befriending* interventions to reduce loneliness. The former category involves facilitating interaction between peers, whereas the latter focuses on actively making new friendships. The stronger evidence for *Social facilitation* found in this review suggests that providing a means for isolated or lonely people to interact with their existing social circles may be more beneficial than making new friends. However, these findings should be interpreted with caution as few studies on befriending interventions were identified. Future high-quality randomised studies of befriending, and in particular telephone befriending, are required to further evaluate its effectiveness.

It is generally accepted that the COVID-19 pandemic has had a disproportionate effect on vulnerable groups, widening pre-existing socioeconomic, race, gender and other inequalities

across the population [7, 78, 79]. Most of the studies reported in this review were found to target older adults, either in the community or in residential, nursing and care homes. Loneliness and social isolation within nursing and care homes has received particular attention due to policies prohibiting family visits and social gatherings at these facilities due to COVID-19 [80, 81]. We found evidence in support of cognitive/psychological interventions, videoconferencing, Wii gaming, gardening and robotic pets as effective interventions in these settings.

Growing evidence suggests that women, ethnic minorities, young adults, and people with lower education or income are at a significantly increased risk of being lonely as a result of the pandemic [82, 83]. We found few studies aimed at young adult or student populations, who may be more vulnerable to loneliness if isolating away from home for prolonged periods. All included studies in this age group involved a psychological or cognitive component, with Mindfulness-based and cognitive-behavioural therapies proving effective. Whether the other categories of interventions identified in this review are similarly effective among young adults is not known. Likewise, very few interventions were identified that specifically target individuals of lower socioeconomic status or ethnic minorities.

Many of the effective interventions in this review will require telephone or video call technology to carry out the intervention during COVID-19 shielding measures. This has implications for the accessibility of each intervention: the costs of the technology required to deliver interventions may restrict participants by socioeconomic status, while the minimum level of digital literacy required may prevent its use among people with lower education [84, 85]. There is a considerable risk that those who are most likely to be lonely or isolated—and hence most in need of interventions—will not possess, or know how to use, electronic devices and/or a high-speed internet connection to facilitate intervention delivery. Any approach to help people suffering from loneliness or social isolation must therefore take these issues into consideration.

Since starting this review, the UK Government has announced a £5 million Loneliness COVID-19 Grant Fund for national organisations working to tackle loneliness [86]. This aims to support charities currently offering services such as telephone befriending and community volunteering schemes [87, 88]. In addition, the NHS.uk website provides both support for people feeling lonely and onward referral for psychological therapies if appropriate [89]. This review expands on the current provision of available services for lonely or isolated individuals and presents the evidence for alternative interventions that comply with COVID-19 distancing measures. We believe a combination of educational and psychological approaches that target the root cause of one's loneliness, in addition to social facilitation initiatives to create and maintain relationships, represent the best opportunities to improve loneliness. It is imperative that researchers and policymakers work together to develop safe, effective programmes that alleviate loneliness and social isolation, while simultaneously addressing the digital, socioeconomic and generational inequalities that may result from unequal access to interventions.

## Strengths

One strength of our analysis is the use of official March 2020 UK government guidance on shielding. This provided an objective method by which to assess the feasibility of interventions. Similar guidance is in place worldwide, so our findings are likely generalizable to other countries. Due to the changing severity of government distancing regulations, we focused on interventions deemed feasible, some with modification, under the most stringent restrictions. Feasible interventions can therefore be conducted irrespective of future, more lenient changes to government policy. Many interventions could also be delivered without modification as restrictions are eased. Moreover, we followed established guidance on the conduct of rapid

reviews, performing a systematic review of systematic reviews to generate the final list of primary studies to be screened. This method allowed a broad and comprehensive review of the existing literature and enabled large numbers of potentially relevant studies to be identified. However, as we were dependent on the search strategy and selection criteria of the reviews identified by our initial search, some relevant studies not reported in a review may have been omitted. We sought to mitigate this by searching the pre-print archive MedRxiv for the most recently published studies.

## Limitations

Our review has several limitations. First, many studies were found to be of "Fair" quality when assessing risk of bias. This was generally because studies did not adequately account for participant loss to follow up, while the nature of many mental health-related interventions means blinding is often not possible. Second, the extent to which our findings can be applied to the entire population is unclear. The country and setting in which interventions were carried out varied, while older adults were the target participants of most studies. Whether the interventions included in this study are similarly effective in younger age groups is not known. It is of paramount importance that effective interventions targeting each age group across different settings are available. Third, there is much to discover about the types of loneliness across different groups affected by the pandemic and ensuing lockdown. Greater understanding of the differences between these groups, and the underlying processes driving various states of well-being, would provide a better foundation to develop interventions that treat loneliness and social isolation for all.

## Conclusion

In conclusion, this review presents the current evidence for interventions targeting social isolation or loneliness that may be compatible with shielding/social distancing measures. Most effective interventions for loneliness either involved cognitive or educational components, or facilitated communication and networking between peers; we found few effective interventions for social isolation. Delivery of available interventions may require modification to align with COVID-19 shielding/social distancing measures—many interventions involved physical contact in their original protocol but were deemed feasible using telephone or video call technology. This has implications for the accessibility of interventions to the wider public. Future high-quality randomised controlled trials conducted under the constraints of shielding/social distancing are urgently needed to build on the findings of this review.

## Supporting information

**S1 File.**
(DOCX)

## Author Contributions

**Conceptualization:** Christopher Y. K. Williams.

**Data curation:** Christopher Y. K. Williams.

**Formal analysis:** Christopher Y. K. Williams, Alice F. Ferreira, Julieta Galante.

**Investigation:** Christopher Y. K. Williams, Adam T. Townson, Milan Kapur, Alice F. Ferreira, Rebecca Nunn, Veronica Phillips.

**Methodology:** Christopher Y. K. Williams, Julieta Galante, Veronica Phillips, Juliet A. Usher-Smith.

**Project administration:** Christopher Y. K. Williams.

**Supervision:** Christopher Y. K. Williams, Julieta Galante, Sarah Gentry, Juliet A. Usher-Smith.

**Validation:** Christopher Y. K. Williams, Adam T. Townson, Milan Kapur, Alice F. Ferreira.

**Visualization:** Christopher Y. K. Williams, Adam T. Townson, Alice F. Ferreira, Julieta Galante, Juliet A. Usher-Smith.

**Writing – original draft:** Christopher Y. K. Williams.

**Writing – review & editing:** Christopher Y. K. Williams, Adam T. Townson, Milan Kapur, Alice F. Ferreira, Rebecca Nunn, Julieta Galante, Veronica Phillips, Sarah Gentry, Juliet A. Usher-Smith.

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
