## [Decision Letter · Decision Letter 0]

14 Oct 2020

PONE-D-20-25726

Interventions to reduce social isolation and loneliness during COVID-19 physical distancing measures: A rapid systematic review

PLOS ONE

Dear Dr. Williams,

Thank you for submitting your manuscript to PLOS ONE. After careful consideration, we feel that it has merit but does not fully meet PLOS ONE’s publication criteria as it currently stands. Therefore, we invite you to submit a revised version of the manuscript that addresses the points raised during the review process.

We look forward to receiving your revised manuscript.

Kind regards,

Cindy Gray, Ph.D.

Academic Editor

PLOS ONE

Journal Requirements:

Additional Editor Comments:

This is an interesting timely and well conducted review.

The abstract is well written and comprehensive.

Introduction: it might be useful to update this to the current situation with regard to COVID-19 in the UK and internationally particularly around restrictions being rei-mposed.

Methods: Well described and seems robust. The categories of loneliness reported are justified. Exclusions are justified. One limitation is that more recent studies - those not included in a review - will not be captured by the methodology. This decision (to do a rapid review) should be more clearly justified In the method. Narrative synthesis is justified.

Results: Would it also be worth thinking about interventions that would be feasible under strict covert measures, but not quite shielding, which seems to be the current situation in the UK at the moment? I think this would just require another section/ paragraph in the results. It would be good to know which categories of Gardiner's framework the non-viable and unclear interventions were in. It would have been good to look at the effectiveness of the programmes on different populations; for example, older people who may be particularly vulnerable and at risk during COVID-19 restrictions - there is an oblique reference to this in the discussion (second paragraph on page 16) but it should be made more explicit that older adults are a vulnerable category.

Discussion: Rather than interventions that target the whole age population, would it not be better to think about future interventions that are targeted at specific age groups?

Reviewers' comments:

Reviewer's Responses to Questions

**Comments to the Author**

1. Is the manuscript technically sound, and do the data support the conclusions?

Reviewer #1: Yes

2. Has the statistical analysis been performed appropriately and rigorously? 

Reviewer #1: Yes

3. Have the authors made all data underlying the findings in their manuscript fully available?

Reviewer #1: Yes

4. Is the manuscript presented in an intelligible fashion and written in standard English?

Reviewer #1: Yes

5. Review Comments to the Author

Reviewer #1: Many thanks to the authors for this submission. This research offers a timely and original contribution to the rapidly emerging Covid-19 field of research.

The rationale and methodological approaches are transparent and have been expertly managed to produce the review.

The findings and implications of the rapid review are of relevance across a number of professional disciplines as well as for policy and practice in helping to inform responses which offer evidence based interventions to reduce social isolation at this time.

Two additional aspects to the review could be considered. One is touched upon on line 357. 1) Addressing inequality in access/inclusion, inclusive of digital literacy could be more fully considered. A couple of ways this would be addressed is by highlighting if any of the reviewed studies provide a blueprint/demonstrate good practice in the area of accessibility for groups at risk of inequality in access such as older age adults - many of whom are facing the highest levels of social isolation (previously shielding). Some references to those populations who are facing highest levels of social isolation and loneliness would be valuable to evidence, so thought can be given to the design/tailoring to the populations affected. At the moment, the review is perhaps quite homogeneous in how it considers social distancing/covid related measures on 'people'. However as said previously, we know that this pandemic is reinforcing inequalities within specific populations such as those on low income, older age adults, BME groups (cultural competency and representations of mental health may be relevant to mention). It is possible to pull out a bit more about the participant/population characteristics from the review? Furthermore, 2) it may be valuable to align some of the findings to existing social support which has been offered through Government_ NHS provision during Covid-19 period thus far. A quick check indicates that these are broadly psychoeducational, befriending in local community and psychological referral. It would be valuable to make a link in discussion which situates the findings of the review within a comment to current provision and how this provision can be improved, re-orientated to take into account the review findings about efficacy of interventions. Many thanks again for submitting this research article.

6. PLOS authors have the option to publish the peer review history of their article (what does this mean?). If published, this will include your full peer review and any attached files.

Reviewer #1: **Yes: **Dr Kate Reid

---

## [Author Response · Author response to Decision Letter 0]

18 Nov 2020

Response to reviewers

Authors: We would like to thank the academic editor and reviewer for their helpful comments. Our detailed point-by-point responses are provided below.

Editor 1:

Editor: This is an interesting timely and well conducted review.

Authors: Many thanks for your kind comments.

Editor: The abstract is well written and comprehensive.

Editor: Introduction: it might be useful to update this to the current situation with regard to COVID-19 in the UK and internationally particularly around restrictions being reimposed.

Authors: We agree. We have updated lines 75-77 (of the tracked-changes manuscript) to reflect the re-introduction of national lockdowns.

Editor: Methods: Well described and seems robust. The categories of loneliness reported are justified. Exclusions are justified. One limitation is that more recent studies - those not included in a review - will not be captured by the methodology. This decision (to do a rapid review) should be more clearly justified In the method. Narrative synthesis is justified.

Authors: We agree. We have clarified the need to conduct a rapid review to urgently inform healthcare policy decisions in lines 105-106. Furthermore, we have acknowledged as a limitation in the Discussion that some recent studies may not have been captured and explained how we sought to mitigate this.

Editor: Results: Would it also be worth thinking about interventions that would be feasible under strict covert measures, but not quite shielding, which seems to be the current situation in the UK at the moment? I think this would just require another section/ paragraph in the results. 

Authors: Within the 58 studies, we have already included all the interventions which do not entirely meet the shielding measures, but would be feasible with modification (these were classed as “Potentially Feasible”, and required modification to bring them in alignment with the principles of shielding). We feel that with the ongoing changes to government COVID-19 restrictions, a standardised approach to classify interventions using the original definition of shielding is most appropriate. Included interventions will therefore be feasible regardless of the current severity of restrictions and will remain feasible as these restrictions change in the future.

Editor: It would be good to know which categories of Gardiner's framework the non-viable and unclear interventions were in. 

Authors: We have expanded the paragraph on “Not feasible” or only “Part feasible” interventions to also detail the types of interventions that were excluded. Full details of these interventions can be found in Supplementary Table 1.

Editor: It would have been good to look at the effectiveness of the programmes on different populations; for example, older people who may be particularly vulnerable and at risk during COVID-19 restrictions - there is an oblique reference to this in the discussion (second paragraph on page 16) but it should be made more explicit that older adults are a vulnerable category.

Authors: We are grateful for this suggestion. We have added a subsection to the Results that covers effective interventions for vulnerable groups. Furthermore, in the Discussion we have discussed the current provision for interventions targeting certain vulnerable groups.

Editor: Discussion: Rather than interventions that target the whole age population, would it not be better to think about future interventions that are targeted at specific age groups?

Authors: We agree. We have re-structured the Discussion to consider different age groups.

Reviewer 1:

Reviewer: Many thanks to the authors for this submission. This research offers a timely and original contribution to the rapidly emerging Covid-19 field of research.

Reviewer: The rationale and methodological approaches are transparent and have been expertly managed to produce the review.

Reviewer: The findings and implications of the rapid review are of relevance across a number of professional disciplines as well as for policy and practice in helping to inform responses which offer evidence based interventions to reduce social isolation at this time.

Authors: Many thanks for your kind comments.

Reviewer: Two additional aspects to the review could be considered. One is touched upon on line 357. 1) Addressing inequality in access/inclusion, inclusive of digital literacy could be more fully considered. A couple of ways this would be addressed is by highlighting if any of the reviewed studies provide a blueprint/demonstrate good practice in the area of accessibility for groups at risk of inequality in access such as older age adults - many of whom are facing the highest levels of social isolation (previously shielding). Some references to those populations who are facing highest levels of social isolation and loneliness would be valuable to evidence, so thought can be given to the design/tailoring to the populations affected. At the moment, the review is perhaps quite homogeneous in how it considers social distancing/covid related measures on 'people'. However as said previously, we know that this pandemic is reinforcing inequalities within specific populations such as those on low income, older age adults, BME groups (cultural competency and representations of mental health may be relevant to mention). It is possible to pull out a bit more about the participant/population characteristics from the review? 

Authors: We agree. We have added sections to the Results and Discussion which discusses the availability of interventions for vulnerable groups and highlighted the lack of effective interventions for some of these groups. We were unable to find studies providing a blueprint for increasing the accessibility of interventions, however we have warned of the necessity to consider accessibility when seeking to deliver interventions in the Discussion. Likewise, we have discussed the different types of accessibility (educational, economic and infrastructure) that may be required to facilitate intervention delivery.

Authors: Full details of the participant characteristics for each study have been extracted and can be found in the Supplementary Tables. We have clarified the reference to this in line 231.

Reviewer: Furthermore, 2) it may be valuable to align some of the findings to existing social support which has been offered through Government_ NHS provision during Covid-19 period thus far. A quick check indicates that these are broadly psychoeducational, befriending in local community and psychological referral. It would be valuable to make a link in discussion which situates the findings of the review within a comment to current provision and how this provision can be improved, re-orientated to take into account the review findings about efficacy of interventions. 

Authors: We agree. We have added a section to the Discussion detailing the current efforts and services for loneliness and social isolation and summarising how this review adds to this current provision. We have further highlighted the need for collaboration to improve loneliness and isolation and address the growing inequalities that may be worsened by unequal access to the currently available interventions.

Reviewer: Many thanks again for submitting this research article.

---

## [Decision Letter · Decision Letter 1]

2 Feb 2021

Interventions to reduce social isolation and loneliness during COVID-19 physical distancing measures: A rapid systematic review

PONE-D-20-25726R1

Dear Dr. Williams,

We’re pleased to inform you that your manuscript has been judged scientifically suitable for publication and will be formally accepted for publication once it meets all outstanding technical requirements.

Kind regards,

Cindy Gray, Ph.D.

Academic Editor

PLOS ONE

Additional Editor Comments (optional):

Your revisions have improved what was already a really interesting paper.

I have a few very minor suggestions that I spotted as I read through the revised manuscript. I hope these are helpful as you make the final preparations for publication.

Ln 150-154 – you might want to think about the tense here

Ln 204 should be Figure 1, not Fig 1

Ln 284 I don’t think commas are needed here : these studies, which did report a significant result, was

Ln 436-440 -consider which tenses are best to use here

Ln 445-447 The following sentence is a bit clumsy and this hard to follow “We sought to mitigate this by searching the pre-print archive MedRxiv for the most recent studies, while included reviews were published as recently as 2020.”

Reviewers' comments:

Reviewer's Responses to Questions

**Comments to the Author**

1. If the authors have adequately addressed your comments raised in a previous round of review and you feel that this manuscript is now acceptable for publication, you may indicate that here to bypass the “Comments to the Author” section, enter your conflict of interest statement in the “Confidential to Editor” section, and submit your "Accept" recommendation.

Reviewer #1: All comments have been addressed

2. Is the manuscript technically sound, and do the data support the conclusions?

Reviewer #1: Yes

3. Has the statistical analysis been performed appropriately and rigorously? 

Reviewer #1: Yes

4. Have the authors made all data underlying the findings in their manuscript fully available?

Reviewer #1: Yes

5. Is the manuscript presented in an intelligible fashion and written in standard English?

Reviewer #1: Yes

6. Review Comments to the Author

Reviewer #1: Following peer review, the authors have addressed the comments and suggestions. This paper is now ready for publication. It offers a useful contribution to the field and will no doubt be beneficial to researchers and stakeholders in this area of study.

7. PLOS authors have the option to publish the peer review history of their article (what does this mean?). If published, this will include your full peer review and any attached files.

Reviewer #1: **Yes: **Dr Kate Reid

---

## [Editor Report · Acceptance letter]

8 Feb 2021

PONE-D-20-25726R1 

Interventions to reduce social isolation and loneliness during COVID-19 physical distancing measures: A rapid systematic review 

Dear Dr. Williams:

I'm pleased to inform you that your manuscript has been deemed suitable for publication in PLOS ONE. Congratulations! Your manuscript is now with our production department. 

Kind regards, 

on behalf of

Dr. Cindy Gray 

Academic Editor

PLOS ONE